# New Insights into Cellular Impacts of Metals in Aquatic Animals

**Aimie Le Saux [1], Elise David [2], Stéphane Betoulle [2], Florence Bultelle [1], Béatrice Rocher [1], Iris Barjhoux [2] and Claudia Cosio [2,\***

[1] Normandie Univ , UNIHAVRE, UMR I-02 SEBIO, FR CNRS 3730 SCALE, 76600 Le Havre, France; aimie.le-saux@univ-lehavre.fr (A.L.-S.); florence.bultelle@univ-lehavre.fr (F.B.); beatrice.rocher@univ-lehavre.fr (B.R.)

[2] Université de Reims Champagne Ardenne, URCA, UMR I-02 SEBIO, 51097 Reims, France; elise.david@univ-reims.fr (E.D.); stephane.betoulle@univ-reims.fr (S.B.); iris.barjhoux@univ-reims.fr (I.B.)

\* Correspondence: claudia.cosio@univ-reims.fr

**Abstract:** Toxic metals remain a current important threat to aquatic ecosystems, despite regulatory efforts to reduce their release. Several toxic metals already appear in the list of priority substances polluting surface waters, while concerns arise from the increasing use of technology-critical metals such as metallic nanoparticles, rare-earth, and platinum group metals. In aquatic environments, various chemical, biological and physical processes determine the impact of metals on the biota. This review provides insights into responses to toxic metals recently reported in freshwater and marine animals. The specific emphasis is on: (i) common cellular and molecular responses; (ii) stress proteins; (iii) redox homeostasis; (iv) cytoskeleton rearrangement; (v) metabolism reshuffle; (vi) free cellular energy and mitochondrial metabolism; and (vi) immunity. These endpoints are promising, notably in multi-biomarker approaches to identify precise cellular toxicity pathways and anticipate the impact of environmental metal pollution.

**Keywords:** bivalve; cytoskeleton rearrangement; free cellular energy and mitochondrial metabolism; metabolism reshuffle; redox homeostasis; stress protein

## 1. Introduction

Although considerable efforts for reducing pollutant release have been made, contamination in surface waters is still of concern. In this context, a deeper understanding of contaminant flux, dynamics, and impact on biota in aquatic systems are needed to better predict their response to changing conditions and efficiently anticipate risks for ecosystems. Nearly all water bodies are contaminated to some level with transition metals, such as cadmium (Cd), lead (Pb), and mercury (Hg), due to anthropogenic sources [1,2]. Transition metals may or may not have an essential biological role. For example, copper (Cu) is essential to all animals as it participates in fundamental physiological processes (e.g., mitochondrial respiration) and is a cofactor for many enzymes, e.g., superoxide dismutases (SOD) and cytochrome c oxidases (CCOX) [3]. Because of its high reactivity, Cu concentration is tightly regulated inside cells by a complex homeostasis network [4]. This homeostasis network has been studied in several model species and there is evidence of high conservation throughout evolution [5,6]. However, Cu in excessive concentrations causes oxidative stress due to adverse effects on the above-mentioned necessary cellular processes, such as enzyme activity and the electron transport chain [7–9]. Similarly, nonessential metals such as Cd are also highly reactive with cellular components and are toxic to cells at low concentrations. Nonessential metals are generally taken up through essential metal homeostasis networks. In different species, metals will bioaccumulate variably in different organs. Moreover, some

metals are more toxic than others in different species. For example, mussels seem more sensitive to Cu than to Cd [10].

Metal concentrations are consequently an important parameter for environmental quality in natural environments. Elevated metal concentrations in aquatic ecosystems are directly related to human activities involving the production of industrial (e.g., pesticide use and agricultural run-off, mine tailings) and domestic wastes (e.g., urbanization, automobile exhausts). In pristine and lightly contaminated systems, the total metal concentration ranges are 100–500 ng Cu/L, 30–200 ng Zn/L, 10–70 ng Cd/L, and 0.05–4.0 ng Hg/L, but total concentrations in oxic surface waters can easily reach 2000 ng Cu/L, 560 ng Zn/L, 350 ng Cd/L and 100 ng Hg/L and higher in severely contaminated sites. However, many toxic metals and metalloids that enter aquatic systems are absorbed into suspended particles or dissolved organic matter prompting their accumulation in surficial sediments and reducing their bioavailability [11,12]. Bioavailability is the proportion of total metals that can be incorporated into biota (bioaccumulation). Knowledge of geochemical parameters in both water and sediment is necessary to understand controls on metal bioavailability and subsequent toxicity in natural waters. Bioavailability studies indicate that aquatic organisms uptake free metal ions (metal hydroxides) from solution quite efficiently and models such as the Biotic Ligand Model were developed to estimate metal bioavailability and subsequent toxicity [13]. Nonetheless, some metals (e.g., Hg) and many biological factors impacting metal bioaccumulation in aquatic organisms in natural waters are not well modeled, limiting the prediction of metal toxicity [13]. However, acute toxic concentrations of dissolved metals resulting in obvious and rapid systemic toxicity (i.e., on survival, growth, and/or reproduction) are uncommon in oxic surface waters. Nevertheless, these metals in environmental concentrations are bioaccumulated and can directly affect aquatic species [14,15].

Ecotoxicology aims to study the effects of toxic substances on the health of ecological systems and their constituent species. In the environment, toxic substances generate adverse effects at all levels of biological organization, from the molecular level to communities and ecosystems. It is hypothesized, for example, in the adverse outcome pathway theory, that cascading chains of causal events occur at different levels of biological organization, i.e., molecular events producing a measurable ecotoxicological effect. In this context, current research priorities in ecotoxicology are to understand mechanisms by which contaminants disturb normal biological performance at a sublethal level relevant in chronic exposure scenarios, and to use this knowledge to develop new early-warning biomarkers.

In recent years, early, specific and robust biomarkers directly linking responses at cellular levels to the whole organism, population, and community-level effects have been envisioned. The development of multi-biomarker strategies that allow the simultaneous monitoring of a wide range of biological responses is seen as useful for improving environmental hazard assessment in widespread non-model sentinel species to generalize the use of these techniques in field applications. Integrating data obtained on different levels of the cellular cascade triggered by a pollutant gives a more complete picture of the health status of an organism. In this context, this paper reviews a set of cellular biomarkers that are seen as promising to assess early cellular toxic effects in aquatic animals, in light of recent researches.

## 2. A Common Response to Toxic Metal Exposure

Once metal toxicity is established, a similar chain of events linked to oxidative stress sets in motion at the subcellular level: the formation of metal-rich granules and the increased production of free radicals, either strictly by the direct effect of a reactant with the metal or by the indirect response to an increase in metal cellular concentration. The production of reactive oxygen species (ROS) has several cellular effects on aquatic species. For example, Cu generates free radicals through the Fenton and Haber-Weis reaction:

$$Cu(II) + O_2^- \rightarrow Cu(I) + O_2$$

$$Cu(I) + H_2O_2 \rightarrow Cu(II) + \cdot OH + OH^-$$

$$2O_2^- + 2H^+ \rightarrow H_2O_2 + O_2$$

By contrast, Cd, which is not a Fenton metal, generates them indirectly [16]. For example, in copepods, 100 µg/L of Cu for four days trigger a significant production of ROS, but a similar exposure to Cd does not [17]. However, an overproduction of ROS induces redox homeostasis disturbances and macromolecule damage. For instance, Cu and Cd exposure induced DNA damage and hedgehog cells (i.e., heavily DNA damage cells) in medaka larvae [18], likely due to ROS overproduction as shown in rainbow trout [19,20] and/or to the inhibition of DNA repair mechanisms [21–23]. Moreover, an excess of metal ions can cause protein mismetallation resulting in a loss of function of proteins [24], further increasing cellular damage. Metal toxicity (specifically Cd) has recently been reviewed by Zhang et al. [25] and will not be detailed here.

Cellular defense mechanisms against toxic metals include canonical antioxidant systems aimed at capturing and neutralizing ROS and more metal-specific responses like sequestration by metallothioneins or into lysozymes. The latter is related to the cell's ability to detect the presence of metals. Cellular metal overload sensing mainly relies on MTF-1 (metal response transcription factor 1) as recently reviewed by Park et al. [26]. This TF was first described in mammals and more recently identified in aquatic species such as oysters and tilapia [27,28]. Briefly, activation of MTF-1 triggers its nuclear translocation; the TF can then bind to metal response elements and activate downstream gene expression, for instance, metallothionein encoding genes. In addition, cross-regulation between metal-response pathways and heat stress response (HSR) has been highlighted [26], suggesting a synergistic response to metal stress.

In the context of environmental risk assessment, stress response biomarkers can be categorized as (i) overexpression of stress proteins, i.e., metallothioneins and chaperones, (ii) redox homeostasis alteration, (iii) cytoskeleton rearrangement, i.e., toxic effects and resistance effects prompting differences in the cell structure, (iv) disruption of metabolic homeostasis, (v) free cellular energy as well as mitochondrial metabolism, and (vi) immunity modulation.

## 3. Overexpression of Stress Proteins

### 3.1. Metallothioneins

Metallothioneins (MTs) are a highly conserved family of small cysteine-rich proteins implicated in metal binding, such as Zn or Fe, and responses to metal stress (Table 1). These proteins have been described extensively in aquatic species, e.g., marine bivalves [29,30], seawater shrimps *Palaemonetes argentines* [31], copepods [17], freshwater gammarids and teleostean fish [32].

MTs are widely used as an environmental surveillance tool. Their expression is highly correlated with metal concentrations in the medium, although the induction threshold varies according to metals [26,33]. MTs binding capacity can also be modified by different abiotic factors, e.g., temperature [34]. In coastal areas, *Mytilus* sp. has been identified as a model species that reflects a linear relationship between MT production and toxic metal exposure: mussels transplanted to a metal-rich site during eight months, exhibited an increase in MTs content. In digestive glands, MTs and metal concentrations were strongly correlated and remained observable throughout the study, despite seasonal variations [35]. Metal cross-influences have also been studied and a correlation between Zn, Cu, and Cd uptake was demonstrated in bivalves [36]: Zn-exposed oysters *Crassostrea hongkongensis* showed higher bioaccumulation of Cu and Cd. Authors hypothesized that Zn induced MT production that also had a high affinity with other metals resulting in increased polymetallic bioaccumulation.

Most aquatic species display a rise in expression of MTs at the transcriptional and proteic levels when exposed from 30 ng/L to 500 mg/L of Cd and Cu [29,32,34]. For example, freshwater mussels *Dreissena r. bugensis* show a four-fold increase in MTs when exposed to 50 or 100 µg/L of Cd for seven days [37]. *Mytilus* spp. require higher metal concentrations to induce MTs and therefore seem particularly resistant to this metal toxicity. Mussels also have a second defense line against toxic metal poisoning that is biomineralization [38]. In higher vertebrates, the quickest upregulation of MT appears in gastrointestinal tissue, gills, and other organs known to be involved in detoxification processes [32].

Since the aforementioned review, protein expression variations have been found to be species-specific and seem to be linked to the physiological state of the organism [35]. For example, size and season impact the production of MT in exposed freshwater species [32,35].

*3.2. HSP*

Heat shock proteins (HSPs) were first discovered in a thermal regulation experiment context. They form a family of proteins involved in folding, refolding, and remodeling during protein synthesis [10,39]. These proteins have been reviewed in fish and bivalves by Basu et al. [40] and Fabbri et al. [41], respectively. With its inducible and constitutive isoforms, HSP70 is the most studied of the HSP family and has been described in the cytosol, mitochondria, and the endoplasmic reticulum [40]. Depending on the species, HSPs show different responses to toxic metal exposure (Table 1). For instance, HSP60 avoids misfolding whereas HSP70 prevents aggregation in stress situations.

Toxic metals have been shown to bind to HSPs, inhibiting their function and causing protein misfolding. As such, the modulation of HSP's expression reflects the level of damage of tissues in mussels and crustaceans [17,42,43].

In fish, three HSPs have been described in a metal contamination context: HPS50, HSP70, and HSP90. Briefly, HSP50 interacts with the endoplasmic reticulum (ER) and functions in the procollagen helix assembly. HSP90 exhibits cytosolic and nuclear locations with the additional roles of signal transduction and transcriptional activation [40]. The abundance of these HSPs is variable throughout the whole organism. In *Sparus auratus* for instance, intraperitoneal Cd injection (1.25 mg Cd/kg body mass) triggers tissue-specific responses with HSP90 and HSP70 being more abundant only in liver and kidney, respectively, and show no significant variation in gills [44].

HSPs expression as general stress biomarkers has been extensively studied in invertebrates. In oysters, overexpression of *HSP60* and *HSP70* mRNA and proteins were documented in concentrations ranging from 100-500 µg/L of Cd [44]. Primary cell cultures isolated from gill or hepatopancreas were submitted to a short Cd exposure (4 h, 5.6 to 22.4 µg/mL). HSP70 and HSP60 exhibited a similar response pattern with an increased abundance in gill cells but not in hepatopancreas cells, while HSP90 level was not modified in both cell types. One part of this study focused on Cd's impact on cell metabolism and will be described in a forthcoming paragraph. In *Mytilus* spp., an increased abundance of HSP70 and HSP60 were described in presence of 100 µg Cd/mussel for seven days [45] and 10 µg Cu/L for seven days [43], respectively. In small crustaceans, i.e., *Tigriopus japonicus*, *HSP20* and *HSP70* mRNA were overexpressed when exposed to 100 µg/L and 50 µg/L respectively for 96 h to most metals, including Cu and Cd [17]. Moreover, this study highlights a downregulation of the *HSP40* gene when exposed to 100 µg/L of Cd and Cu, a potential biomarker for toxic metals. In this species, Cu-induced overproduction of most *HSP* mRNAs (except the aforementioned *HSP40* gene) and Cd-induced *HSP20*, *HSP70,* and *HSP90* genes and a downregulated *HSP10* gene.

## 4. Redox Homeostasis Disturbance

As mentioned above, metal toxicity induces ROS production. When cell defense capacities are overwhelmed, ROS may trigger oxidative stress. Antioxidant enzymes, such as superoxide dismutases (SOD), catalases (CAT) or antioxidant molecules such as glutathione (GSH), are widely used as molecular markers for metal exposure monitoring, alone or in combination with multi-biomarker approaches (Table 2). Scientists also track macromolecular and cellular damages. In animal cells, the mitochondria are very often analyzed, showing destabilization and function modification due to oxidative stress resulting from toxic metals [16,46]. This destabilization has direct consequences on cellular respiration (see Section 7). At the cellular level, lipid peroxidation and lysosomal activity can be affected by metal toxicity [16,46]. These effects have been reviewed in *Mytilus* spp. [47].

Effects of Cd on reduced glutathione (GSH), a major non-enzymatic compound of antioxidant defense, are well documented and have been reviewed by Nuran et al. [48]. In *Mytilus edulis*, a metal-specific response was identified when exposed to Cu at 40 µg/L for six days, with a GSH

decrease (−25%) in the digestive gland and gills, but no significant variation when exposed to 40 µg/L of Cd or Zn [49]. A decrease in GSH was also observed in *C. virginica* gill and hepatopancreas cell cultures in response to Cd exposure (four hours at 22.4 µg/mL; [50]). By contrast, in the copepod *T. japonicus*, 100 µg/L of Cu during 96 h increased GSH content [17], but 100 µg/L of Cd had no effect. Moreover, in both exposures, increased activities were observed for GST (glutathione S transferases), GR (glutathione reductases), and SOD enzymes. SOD proteins are the first defenses against ROS damages. In *Mytilus* spp., three isoforms have been characterized in response to Cu (25 µg/L for seven days; [47]). Transcriptomic studies on sea bream cell cultures exposed to toxic metals revealed contrasted results: for example, no *SOD* gene expression modification was detected by DNA microarray in the fibroblast cell line (SAF1) following a 24 h exposure to Cu or Cd (15.9 mg/L and 1.12 mg/L, respectively) [46]. *SOD(Mn)* and *SOD(Cu)* gene transcription levels were not significantly modulated in medaka larvae and embryos following exposure to Cd-spiked sediments [21]. Similar results were reported in isolated leukocytes exposed to higher concentrations of Cd but for shorter times (5.6 mg/L for two hours), while *CAT* mRNA expression was enhanced [51]. Indeed, CAT protein has been described as a sensitive biomarker to toxic metals [52]. A three-year field study concerning *Mytilus* spp. issuing from contrasted metal contamination sites seems to confirm CAT protein level robustness [53].

**Table 1.** Summary of studies describing metal impacts on stress proteins, i.e., metallothioneins (MTs) and Heat Shock Proteins (HSPs).

| Species | Metal | Concentration | Duration | Effect | |
|---|---|---|---|---|---|
| *Crassostrea virginica* (cell primary culture) | Cd | 5.6 to 22.4 µg/mL | 4 h | Gill cells: ↑HSP60 and HSP70. HSP70 shows a strong correlation with Cd concentrations. Hepatopancreas cells: higher uptake, but HSP70 was unchanged | [50] |
| *Crassostrea hongkongensis* | Zn, Cd, Cu (field) | n.a. | 2 months (caging) | Zn exposure: ↑Cu and Cd uptake and ↑MT | [36] |
| *Echinogammarus acarinatu, Gammarus balcanicus, Salmo trutta* | Cd, Cu (field) | n.a. | lifetime | Contaminated sites: ↑MT levels Contaminated sites: ↑MT levels Intestine: ↓metal content linked to detoxication abilities | [32] |
| *Mytilus edulis* | Cd, Cu, Zn (field) | n.a. | 8 months (caging) | In digestive gland MT abundance is linearly correlated to metal concentration, but not in gills; Seasonal modulation of MT abundance but metal-induction remains measurable | [35] |
| | Cu, Cd | 10 µg/L, 100 µg/mussel | 7 days | Resistant populations induce more HSP70 than sensitive ones | [45] |
| | Cu | 100 µg/L | 7 days | ↑HSP70 and ↑HSP60 | [43] |
| *Mytilus galloprovincialis* | Ag (NP) | 10 µg/L | 96 h | ↑MTs and ↑*MT10* gene; no effect on *MT20* gene | [29] |
| *Ostrea edulis* | Cd Zn Zn, Cd | 500 µg/L 500 µg/L 500 µg/L each | 7 days | Gills and digestive gland: ↑MT and HSP70 No effect Gills and digestive gland: ↑MT and ↑HSP70 | [34] |
| *Palaemonetes argentinus* | complex metal mixture (field) | n.a. | 96 h (caging) | Cephalothorax: ↑MT correlated significantly to Cd | [31] |
| *Tigriopus japonicus* | Cd, Cu | 5 to 100 µg/L | 96 h | hsp genes HSP20 and HSP70 are upregulated | [17] |
| *Sparus aurata* | Cd | 1.25 mg Cd/kg body mass (injection) | 7 days | Strong ↑HSP70 in the kidney; ↑HSP90 in liver | [44] |

NP, nanoparticles. n.a., non-available data.

**Table 2.** Summary of studies describing metal impacts on redox homeostasis disturbance.

| Species | Metal | Concentration | Duration | Effect | |
|---|---|---|---|---|---|
| *Mytilus edulis* | Cu, Cd, Zn | 40 μg/L | 6 days | ↓GSH in gill and hepatopancreas with Cu or Cd, but not with Zn | [49] |
| | Cu | 25 μg/L | 7 days | ↑SOD activity for the three identified isoforms | [47] |
| *Mytilus galloprovincialis* | Cd (field) | n.a. | lifetime | Gills: ↑CAT activity and high lipoperoxidation at polluted sites | [53] |
| *Sparus aurata* Leukocytes | Cd | 5.6 mg/L | 2 h | ↑*CAT* gene; no *SOD* gene expression modification | [51] |
| Fibroblast cell line SAF1 | Cu, Cd | 15.9 mg/L, 1.12 mg/L | 24 h | No *SOD* gene expression modification | [46] |
| *Tigriopus japonicus* | Cd, Cu | 100 μg/L | 96 h | ↑GST and SOD activities; ↑GSH with Cu treatment but not with Cd | [17] |

n.a., non-available data.

## 5. Cytoskeleton Rearrangement

The cytoskeleton can be defined as proteins that structure the cell through three main networks: microtubules (tubulin), microfilaments (actin), and intermediary filaments (filamin and lamin). The cytoskeleton is implicated in molecule transport and in signaling via vesicular trafficking. Studies have demonstrated cytoskeleton disruption due to metal exposure (Table 3) [10].

In hemocytes isolated from *Mytilus* spp., actin immunostaining revealed a modified cellular distribution of filaments after Cd and Cu exposure, evidencing cytoskeleton disruption [54]. For lower tested concentrations (5.6 mg Cd/L; 3.18 mg Cu/L), the actin cytoskeleton was increased in the perinuclear zone and reduced in the cortical cell areas. At higher tested concentrations (112 mg Cd/L; 12.72 mg Cu/L), microfilaments appeared less developed and cells exhibited a round shape [54]. Whether these effects are direct or indirect is yet to be determined. Indeed, disturbance of the cytoskeleton is mainly documented by proteomic studies that show variations in the abundance of cytoskeletal elements. In fish, for instance, effects of Cu (50 µg/L, three days) on gill proteomes have revealed species characteristic profiles: actin is decreased in common carp and rainbow trout but shows no modification in gibel carp. Furthermore, F-actin capping protein subunit β is more abundant in gills of gibel carp but not in the two other analyzed species [55]. Tissue-specific responses were observed in *M. galloprovincialis* exposed to Cu (10 µg/L, 15 days): actin abundance was reduced in gills and hepatopancreas, while β-tubulin was upregulated only in the latter [56]. In rock oysters, a metal-specific response was described following a four-day exposure to Cd, Cu, Pb or Zn (5, 50, 100 µg/L): actin, for instance, was downregulated by Cu (5 µg/L) and Zn (100 µg/L), upregulated by Pb (100 µg/L), and not affected by Cd [57].

## 6. Metabolism Reshuffle

Metal toxicity increases energy demand requiring higher ATP production [58]. A widely observed cellular response consists of switching from an aerobic to an anaerobic metabolism producing more ATP (Table 4). This can be triggered by the aforementioned oxidative stress, caused directly by redox metals or indirectly by non-redox metals. This metabolism switch seems to be supported by increased glucose levels and decreased glycogen reserves in fish treated for 10 days with 1 mg Cd/L [59]. The maximal aerobic and anaerobic capacities of the first steps of glycolysis can be assessed by the integrative glycolytic flux method [60]. We achieved a laboratory study in freshwater fish revealing a disturbance of aerobic metabolism in juvenile *Rutilus rutilus* exposed to Cu. During this experiment, juvenile fishes were exposed to 0, 10, 50, and 100 µg/L of Cu for seven days. At 50 µg Cu/L, the glycolytic flux measured in white muscle compared to control, revealed a downward trend in anaerobic flux after one day and an upward trend after seven days, while the aerobic flux remained unchanged throughout the whole exposure [61]. These data suggest a potential switch to anaerobic metabolism induced by Cu in juvenile roaches. Concomitantly, an increased expression of the *CCOX1* gene was observed, while adenylate energy charge was maintained (AEC > 0.7) after one day (T1) despite a decrease in ATP concentration [62,63]. After seven days, however, AEC decreased with increasing concentrations of Cu (AEC < 0.7), while the increase in expression of *CCOX1* was lowered compared to T1. Taken together, these observations support that roaches were first able to cope with Cu exposure maintaining their aerobic metabolism. Eventually, after one week of exposure, organisms seem unable to compensate for this stress and anaerobic metabolism was thereafter mobilized (Figure 1) [61]. Nevertheless, ATP production in bivalves has been described as decreased by metals, with 6-phosphofructokinase (PFK)/pyruvate kinase (PK) ratio decreasing when exposed to Cd (50 µg/L) for 10 days [64,65], a mechanism that is in accordance with the ATP/ADP decrease observed in green mussels exposed seven days to Cd (20 µg/L) and/or Cu (50 µg/L) using metabolomic strategy [66]. However, in some species such as clams, ATP production is maintained during Cu (150 µg/L) exposure, probably due to anaerobic metabolism stimulation, in most tissues except hemolymph [67]. Moreover, the oxidation of NADH during electron transport is affected by Cd [37]. In fish, Cd treatment can inhibit Na$^+$/K$^+$-ATPase activity and expression in the major

osmoregulatory organs: gills and kidneys [44]. Cu impacts the early stages of glycolysis and for instance, has been shown to inhibit hexokinase and phosphofructokinase activity in concentrations as low as 10 µg/L in mussels [68]. Interestingly, in mammals, PFK can counteract both Cu and Cd effects on the cytoskeleton [69].

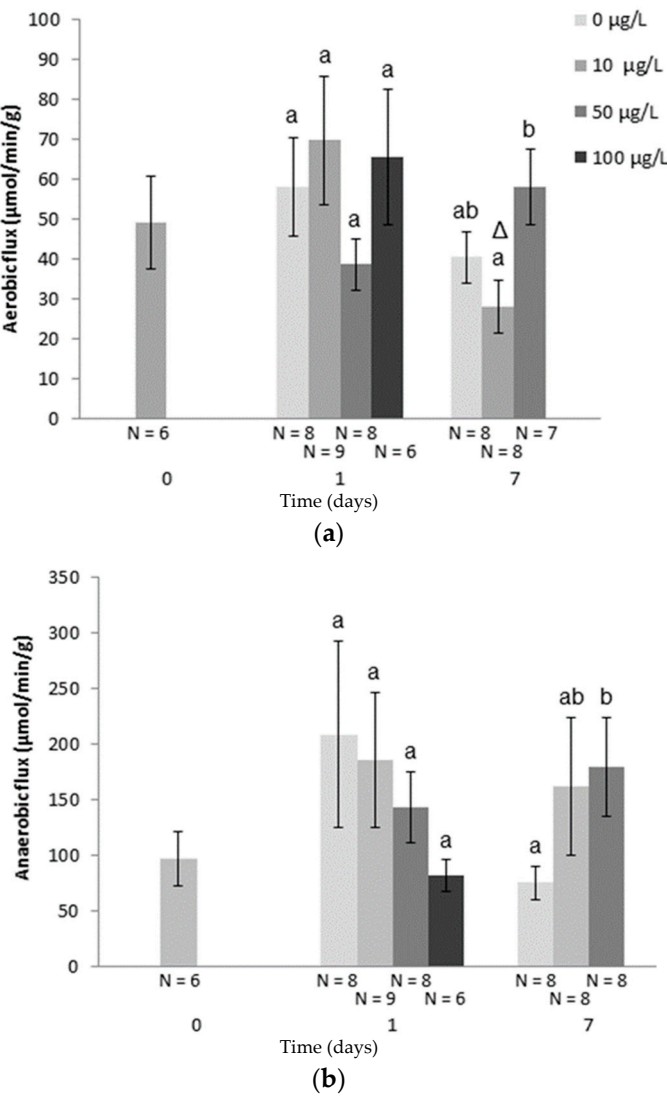

**Figure 1.** (**a**) Aerobic flux (µmol/min/g) and (**b**) anaerobic flux (µmol/min/g) in white muscle of juvenile roach during Cu exposure. Bars represent S.E. Different letters indicate significant differences for the same time point. Δ indicates significant differences for the same concentration among individual samples at T1 and T7.

## 7. Impacts of Metals on Free Cellular Energy and Mitochondrial Metabolism

Energy metabolism provides particularly relevant information about stress responses and tolerance (Table 5). The balance between energy demand and energy supply is critical for the survival of organisms facing environmental stressors such as metal toxicants. Metal contamination increases energetic costs of cell maintenance to ensure protection, detoxification as well as repair (see above), and finally survival [70,71]. The scope for growth measurement (SFG) reflects energy allocation to growth and reproduction. SFG in aquatic organisms shows that metal exposure shifts the use of the energy towards the implementation of defense mechanisms (see references in [72,73]). Indeed, in freshwater mussels *Dreissena polymorpha* exposed to 10 µg Cd/L in controlled conditions, SFG decreased after 90 min [74]. At the cellular level, the mitochondrion is the key organelle for free energy synthesis

as respiratory chain and oxidative phosphorylation provide ATP supply. Mitochondrial electron transport system activity is altered by metal exposure as evidenced by the decrease observed in juvenile roach exposed for one day to 100 μg Cu/L [62]. The activity of respiratory chain enzymes varies depending on the complexes they form. For instance, in rainbow trout mitochondria exposed in vitro to Cu (0–1.3 mg/L), complex I and III basal respiration rates were stimulated while complex II and IV basal respiration rates were inhibited [75]. In mollusks such as marine clams, succinate dehydrogenase (complex II) activity is similarly inhibited in mitochondria in vitro exposed to Cu (5 mg/L) [76]. In eastern oysters, sensitivity and response differences to Cd were also observed between electron transport chain complexes [58]. However, in the same species, metals were shown to induce a decrease of the whole mitochondrial respiration rate [77,78]. Free energy availability is consequently depressed, as reflected by the ATP/ADP ratio and the AEC decline in Cd (25 μg/L; three weeks) exposed oysters [77]. Such a decline of ATP concentration is also observed in oyster hemocytes exposed to Cd (0–22.5 mg/L) [79]. Cd effect on mitochondrial bioenergetics depends on the concentration: an increase of proton leak is observed in oysters' isolated mitochondria exposed to 112.4–562 μg/L of Cd with no effect on ADP-stimulated state 3 respiration rates, while the reverse is observed at 1.12–5.62 mg/L of Cd with the inhibition of state 3 respiration [80]. However, if they are all inhibited in Cd exposed oysters, mitochondrial matrix enzymes, such as citrate synthase or isocitrate-dehydrogenase, are more sensitive to Cd than the membrane enzymes of the respiratory chain complexes [58]. The disturbance of energy metabolism in Cd exposed oysters can differ from their isolated mitochondria, but both support complex phenomenon implying diverse enzymatic impairments [78]. Finally, exposure to a metal such as Cd in oysters, decreases the activation energy of most mitochondrial enzymes, consequently limiting the ability of organisms to face additional environmental stress such as temperature increase [58]. The metabolic effect of metals can also be assessed at the molecular level targeting the same pathways. Indeed, expression of genes involved in cellular energy metabolism, notably in the respiratory chain, may also give relevant information on stress responses. For example, under laboratory conditions, the expression of the *CCOX* gene increased in several species of freshwater and marine mollusks exposed to Cd [81] and in zebra mussels exposed to different metals [82].

## 8. Impacts of Metals on Immunity

Immunity is a physiological system involved in the organism's main homeostatic functions. Its role in anti-pathogen defense (against bacteria, viruses, fungi, etc.) is widely known, but it is also implicated in the tissue remodeling regulation, associated with reproduction, growth, development, and in stress response. Its role in organism integrity maintenance gives it a central position in physiology. Metal toxicity results in immune function modulation in aquatic organisms, involving immune parameter inhibition and/or stimulation of cellular and molecular immunological functions.

Cu increases leucocyte oxidative activity in carp (*Cyprinus carpio*) exposed from four to eight days to metal at environmental concentrations (100 and 250 mg/L in the Champagne region, France) [83]. The humoral immune factor responses of Cu-treated carp were modulated by the presence of a parasite (*Ptychobothrium* sp.), as shown by a high increase in lysozyme activity observed in parasitized carp after exposition [84]. Similar responses were observed in roach (*Rutilus rutilus*) exposed to environmental concentrations of Al [85]. A significant pro-oxidant effect was observed in head kidney leukocytes of roach exposed to 100 μg Al/L for two days. These pro-oxidant effects were higher in fish whose immunity was stimulated with LPS-bacterial endotoxin. These results demonstrate that environmental concentrations of Al induce oxidative and immunotoxic effects in fish and are associated with an immunomodulatory process in relation to inflammatory responses.

Co-exposure of fish to both chemical and biological (parasitic) stress may increase the effects of metals exposure. Identical observations were made in other fish species in laboratory studies [86]. The central question of the environmental reality of these fish responses in their natural environment has rarely been studied to date [87].

## 9. Conclusions

In recent years, responses to toxic metals in aquatic animals were studied in various experimental conditions which helped to confirm that they trigger common toxicity pathways. In this context, various datasets showed that molecular regulation at the gene, protein, and metabolite level indicated physiological adaptation. Metals induce an early response as evidenced by alterations both at structural and functional levels of different organs, including enzymatic and genetic effects. Thus, they affect redox homeostasis, energy metabolism, and the innate immune system of exposed biota and/or potentially increase susceptibility to multiple types of disease (Figure 2). Moreover, the analysis of these endpoints, notably in multi-biomarker strategies, seems highly promising in identifying molecular signatures that will allow the early and sensitive detection of metal contamination. Nevertheless, further work is needed to efficiently use these molecular and cellular responses to enable risk assessment in the environment at population and community levels. To reach this aim, an effort to better link biomarker responses with whole-organism impact has to be a research priority notably to take into account biological variability and the influence of confounding factors [88]. Biomarkers are valuable tools for the implementation of guidelines for effective environmental management as they offer additional biologically and ecologically relevant information. However, biomarker validation in wild populations is necessary, to normalize background levels and eventually to complement standard tests to increase relevant legislation regarding the protection of aquatic environments [89].

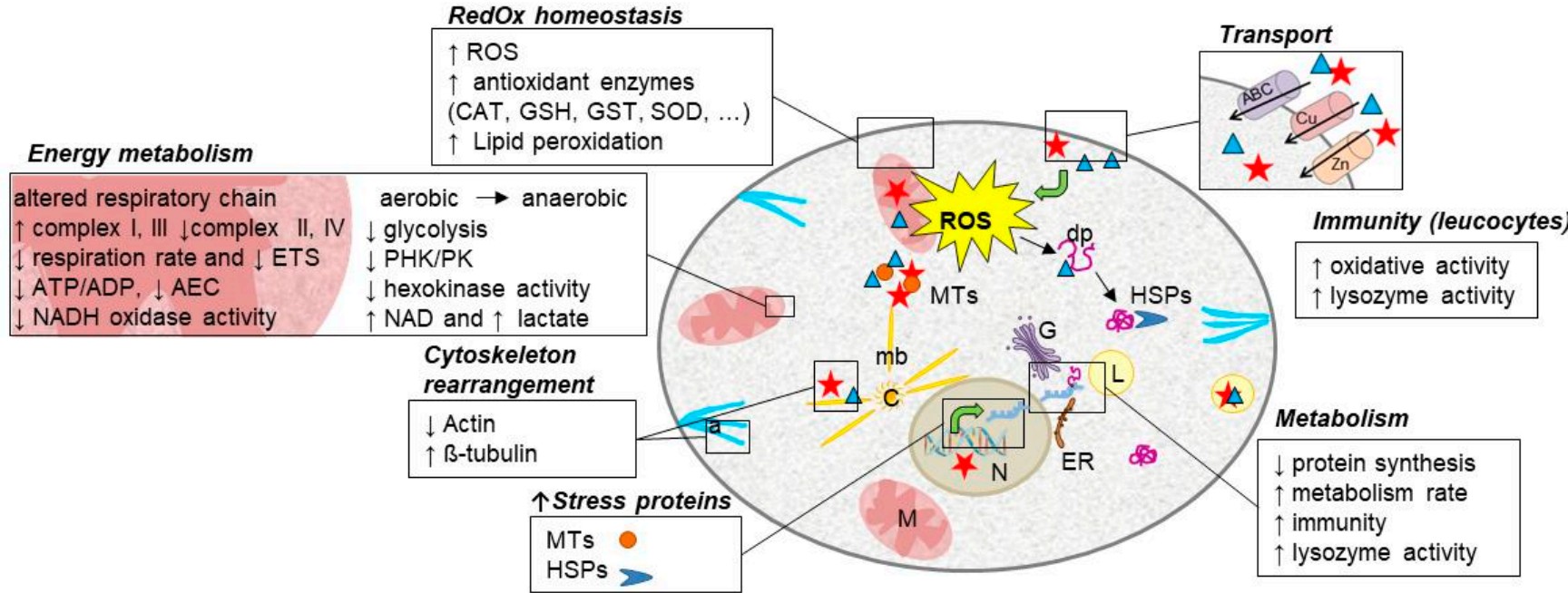

**Figure 2.** Exposure to toxic metals in aquatic animals triggers common toxicity pathways (ABC: ABC transporter; a: actin; C: centrosome; dp: denatured protein; L: lysosome; ER: endoplasmic reticulum; G: Golgi; HSPs: heat shock proteins; M: mitochondria; mb: microtubules; MTs: metallothioneins; N: nucleus; ROS: reactive oxygen species).

**Table 3.** Summary of studies describing metal impacts on cytoskeleton rearrangement.

| Species | Metal | Concentration | Duration | Effect | |
|---|---|---|---|---|---|
| *Carassius auratus gibelio,* *Cyprinus carpio,* *Oncorhynchus mykiss* | Cu | 50 µg/L | 3 days | ↓Actin in common carp and rainbow trout ↑F- actin capping protein subunit β in common carp only | [55] |
| *Mytilus edulis* | Cd, Cu, Pb, Zn, PAHs (field) | n.a. | lifetime | ↑actin carbonylation (gills) and ↑glutathionylation (digestive gland) in polluted sites | [90] |
| *Mytilus galloprovincialis* (cell primary culture) | Cu, Cd | 5.6 to 112 mg Cd/L; 3.18 to 12.72 mg Cu /L | 24 h | Hemocytes: actin cytoskeleton alteration | [54] |
| | Cu | 10 µg/L | 15 days | Gills: ↓actin; Digestive gland: ↓actin, ↑tubulin | [56] |
| *Saccostrea glomerata* | Cd, Cu, Pb, Zn | 5, 50, 100 µg/L | 4 days | ↓Actin (Cu and Zn), ↑actin (Pb) or nor modified (Cd) | [57] |

n.a., non-available data.

**Table 4.** Summary of studies describing toxic metal impacts on metabolism reshuffle.

| Species | Metal | Concentration | Duration | Effect | |
|---|---|---|---|---|---|
| *Crassostrea virginica* | Cd | 22 mg/L | 4 h | ↑Metabolism rate | [58] |
| *Cyprinus carpio* | Cd | 1 mg/L | 10 days | ↓Glycogen levels | [59] |
| *Dreissena bugensis* | Cd | 100 µg/L | 7 days | NADH oxidase activity is negatively correlated with the accumulation | [37] |
| *Mytilus galloprovincialis* | Cu | 80 µg/L | 7 days | ↓Protein synthesis | [65] |
| | Cd, Cu | 40 mg/L | 3 days | ↓hexokinase activity by 35% (Cu) | [68] |
| *Perna viridis* | Cd, Cu | 50 µg/L, 20 µg/L respectively | 7 days | Both metals alone or in combination ↑glycogen, ↑NAD, ↑lactate and ↓ATP/ADP | [66] |
| *Sparus aurata* | Cd | 1.25 mg Cd/kg body mass (injection) | 7 days | ↓activity and expression of $Na^+/K^+$-ATPase in the major osmoregulatory organs, gill, and kidney | [44] |
| *Tegillarca granosa* | Cd | 50 µg/L | 10 days | ↓PHK and ↓PK activities to 70% of the control | [64] |

n.a., non-available data.

**Table 5.** Summary of studies describing toxic metal impacts on free cellular energy and mitochondrial metabolism.

| Species | Metal | Concentrations | Duration | Effect | |
|---|---|---|---|---|---|
| *Crassostrea virginica* | Cd | 45 mg/L | Assay duration | Activity of mitochondrial complexes I to IV: ↓ or ↑ depending on the complex (gills and hepatopancreas) | [58] |
| *Crassostrea virginica* | Cd | 50 μg/L | 30 days | Gills mitochondria: ↓Mitochondria ADP-stimulated respiration rate | [78] |
| *Crassostrea virginica* | Cd | 25 μg/L | 21 days | Gills mitochondria: ↓Mitochondria ADP-stimulated respiration rate<br>Gills: ↓ATP/ADP; ↓AEC | [77] |
| *Crassostrea virginica*<br>*In vitro* exposed mitochondria | Cd | 5,62 mg/L | Assay duration | ↓Mitochondrial respiration rates (gills mitochondria) | [80] |
| *Crassostrea virginica*<br>*In vitro* exposed hemocytes | Cd | 22.5 mg/L | 72 h | ↓ATP | [79] |
| *Dreissena polymorpha* | Cd | 10 μg/L | 7 days | ↓SFG after 90 min<br>↑SFG after 7 days | [74] |
| *Mesodesma mactroides*<br>*In vitro* exposed mitochondria | Cu | 5 mg/L | 1 h | ↓Succinate dehydrogenase activity (gills and digestive gland mitochondria) | [76] |
| *Oncorhynchus mykiss*<br>*In vitro* exposed mitochondria | Cu | 0.32, 0.64, 1.27 mg/L | Assay duration | Liver mitochondria respiratory chain complex:<br>↑Complex I and III basal respiration rates<br>↓Complex II and IV basal respiration rates | [75] |
| *Rutilus rutilus* | Cu | 100 μg/L | 7 days | ↓ETS after 24h (white muscle) | [62] |

**Author Contributions:** Writing—review A.L.S., E.D., S.B., F.B., B.R., I.B.; writing—review and editing C.C.; All authors have read and agreed to the published version of the manuscript.

**Conflicts of Interest:** The authors declare no conflict of interest.

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
