# Peer review of "New Insights into Cellular Impacts of Metals in Aquatic Animals"

_environments, doi:10.3390/environments7060046_

Round 1

Reviewer 1 Report

The paper is a nicely balanced review, although some elements of meta-analysis would add to its value. Some points to consider:

  1. ll. 45ff: The concept of “bioavailability” should be discussed more extensively.

  2. ll. 145-147: This fragment makes one wonder what is the indicator of what? Toxicity/resistance of overexpressed proteins or vice versa? This should be put more clearly.

  3. The Authors should consistently use capital letters in the names of Heat Shock Proteins (HSPs). Similar inconsistency has ocurred in the case of superoxide dismutases (SOD) and catalases (CAT).

  4. Arrows indicating increases or decreases in the tables should be made more conspicuous.

  5. Figures are too small.

  6. Minor remarks concerning language:

    1. Spell out one-digit numbers, e.g. “one” instead of “1”.

    2. Use hyphens in compound terms, e.g. “cysteine-rich” (l. 110), or “Cu-induced” (l. 167).

Some more remarks have been marked directly on the manuscript – see attached file.

Author Response

Reviewer 1

Comments and Suggestions for Authors

The paper is a nicely balanced review, although some elements of meta-analysis would add to its value. Some points to consider:

  1. ll. 45ff: The concept of “bioavailability” should be discussed more extensively.

ANSWER: we added  a short part on the bioavailability and BLM.

Line 52: “However, many toxic metals and metalloids that enter aquatic systems are absorbed into suspended particles or dissolved organic matter prompting their accumulation in surficial sediments and reducing their bioavailability [11, 12]. Bioavailability is the proportion of total metals that can be incorporated into biota (bioaccumulation). Knowledge of geochemical parameters in both water and sediment is necessary to understand controls on metal bioavailability and subsequent toxicity in natural waters. Bioavailability studies indicate that aquatic organisms uptake free metal ions (metal hydroxides) from solution quite efficiently and models such as the Biotic Ligand Model were developed to estimate metal bioavailability and subsequent toxicity [13]. Nonetheless, some metals (e.g. Hg) and many biological factors impacting metal bioaccumulation in aquatic organisms in natural waters are not well modeled, limiting the prediction of metal toxicity [13]. However, acute toxic concentrations of dissolved metals resulting in obvious and rapid systemic toxicity (i.e. on survival,  growth, and/or reproduction) are uncommon in oxic surface waters. Nevertheless, these metals in environmental concentrations are bioaccumulated by biota and can directly affect aquatic species [14, 15].”

  1. ll. 145-147: This fragment makes one wonder what is the indicator of what? Toxicity/resistance of overexpressed proteins or vice versa? This should be put more clearly.

ANSWER: The fragment was deleted and information was reorganized to enhance clarity.

  1. The Authors should consistently use capital letters in the names of Heat Shock Proteins (HSPs). Similar inconsistency has ocurred in the case of superoxide dismutases (SOD) and catalases (CAT).

ANSWER: Initially we used capital letters for proteins and minuscule letters for genes. We modified the text and now use italics for genes and explicitly mention gene/mRNA for clarity.

  1. Arrows indicating increases or decreases in the tables should be made more conspicuous.

ANSWER: This has been done

  1. Figures are too small.

ANSWER: The size of figures has been increased

  1. Minor remarks concerning language:
    1. Spell out one-digit numbers, e.g. “one” instead of “1”.

ANSWER: This has been done throughout the text.

    1. Use hyphens in compound terms, e.g. “cysteine-rich” (l. 110), or “Cu-induced” (l. 167).

ANSWER: This has been done

Some more remarks have been marked directly on the manuscript – see attached file.

ANSWER: This has been done

Reviewer 2 Report

Overview

The scientifically-rigorous impact assessment of contaminants, including heavy metals, in aquatic systems had its genesis in the 1950 and ‘60s. Thus, this field has a very long and deeply-comprehensive literature, with input from intellectual giants from all areas of scholarly endeavours. Such diverse and trans-disciplinary knowledge has combined to produce research and policies that have forever changed the world of ecotoxicology, environmental toxicology, and water pollution research. But as we move forward in the arena of risk assessment and in implementing sustainable environmental protection strategies, we must be careful not to oversell the value of each research effort. Johnson and Sumpter (2016), in their seminal paper report the following: “A very high number of published studies now report changes in gene expression, protein profiles, and metabolite profiles in a variety of aquatic organisms exposed to chemicals, without linking them to any phenotypic change relevant to the animal. We appear to have become very caught up in detail and to have lost sight of the wider picture”.

Johnson, A.C. and J.P. Sumpter. 2016. Are we going about chemical risk assessment for the aquatic environment the wrong way? Environmental Toxicology and Chemistry. 35(7): 1609-1616

Additionally, the history of the evaluation, characterization, and use of biomarkers in ecotoxicology has a long history, with comprehensive research found in the literature, from over 3 decades ago. I refer the authors to an excellent review by Lopez-Barea (1995): Biomarkers in Ecotoxicology: an overview. Archives of Toxicology. 17:57-79

Thus, I frame my evaluation of the current paper in light of the above.

Strengths of the Manuscript:

In the extensive area of water pollution research, the Review on current research (past 15 years) in the area of environmental impact of metals on genomics, proteomics, and metabolomics of aquatic organisms is useful. Indeed, the 88 references bespeaks a nicely-mined analysis of the existing research in this area. In general, the Manuscript adds knowledge to the understanding of complex cellular biology when an organism is exposed to an anthropogenic stressor such as heavy metals. While production of reactive oxygen species and metallothionein expression in organisms exposed to heavy metals have been extensively reported in past research, the Manuscript provided nice details on cellular function pertaining to cytoskeleton and metabolism disruption, and impact to the immune system which was fresh and interesting.

Limitations:

A common flaw in more recent environmental scientific research is the lack of depth in mining the literature to position current research alongside foundational, broader knowledge. Without detailing some of the extensive history of biomarker research in the field of ecotoxicology prior to the past 15 years (as presented in this Manuscript), nor to examining any correlation between contaminant exposure and subsequent whole-organism impact, the Manuscript has made the same mistake. Detecting exposure of biological systems to contaminants through the use of biomarkers has been the focus of study by many researchers for several decades, including such biomarker research as MFO induction and vitellogenin production in male fish. However, correlating exposure of biota to an observable whole-organism impact, with potential subsequent population changes, has not been attempted in this Manuscript and such a limitation should be addressed, even if just briefly. A more nuanced assessment of conventional ecotoxicity tests that monitor observable whole-organism endpoints such as mortality, growth, and reproduction, alongside biomarker research, would add scientific weight to the overall biomarker results and conclusions. While such information is not essential in the current Manuscript, it is important that the authors do not oversell the value of using "multibiomarker approaches" when attempting to protect organisms in the environment, or in conducting relevant environmental risk assessment. It must be remembered that biomarkers can indicate "exposure" but rarely have been correlated with whole-organism impact. This dearth of knowledge must be emphasized in the Conclusions.

Author Response

ANSWER: Our aim was to report recent findings on cellular toxicity (see lines 81-83), but we completely agree with these points and modified the text to address these limitations explicitly. This is now written more in detail line 83. We also added sentences to briefly mention historical ecotoxicology approaches and point limits of current knowledge and data.

Line 62: “However, acute toxic concentrations of dissolved metals resulting in obvious and rapid systemic toxicity (i.e. on survival,  growth, and/or reproduction) are uncommon in oxic surface waters.”

Line 81: “In this context, this paper reviews a set of cellular biomarkers that are seen as promising to assess early cellular toxic effects in aquatic animals, in light of recent researches.”

Line 351: “To reach this aim, an effort to better link biomarker responses with whole-organism impact has to be a research priority notably to take into account biological variability and the influence of confounding factors [88].”